# An improved two-stage label propagation algorithm based on LeaderRank

Miaomiao Liu[1,2], Jinyun Yang[1], Jingfeng Guo[3], Jing Chen[3] and Yongsheng Zhang[1]

[1] School of Computer and Information Technology, Northeast Petroleum University, Daqing, Heilongjiang, China
[2] Key Laboratory of Petroleum Big Data and Intelligent Analysis of Heilongjiang Province, Northeast Petroleum University, Daqing, Heilongjiang, China
[3] College of Information Science and Engineering, Yanshan University, Qinhuangdao, Hebei, China



## ABSTRACT

**Abstract:** To solve the problems of poor stability and low modularity ($Q$) of community division results caused by the randomness of node selection and label update in the traditional label propagation algorithm, an improved two-stage label propagation algorithm based on LeaderRank was proposed in this study. In the first stage, the order of node updating was determined by the participation coefficient (PC). Then, a new similarity measure was defined to improve the label selection mechanism so as to solve the problem of label oscillation caused by multiple labels of the node with the most similarity to the node. Moreover, the influence of the nodes was comprehensively used to find the initial community structure. In the second stage, the rough communities obtained in the first stage were regarded as nodes, and their merging sequence was determined by the PC. Next, the non-weak community and the community with the largest number of connected edges were combined. Finally, the community structure was further optimized to improve the modularity so as to obtain the final partition result. Experiments were performed on nine classic realistic networks and 19 artificial datasets with different scales, complexities, and densities. The modularity and normalized mutual information (NMI) were used as evaluation indexes for comparing the improved algorithm with dozens of relevant classic algorithms. The results showed that the proposed algorithm yields superior performance, and the results of community partitioning obtained using the improved algorithm were stable and more accurate than those obtained using other algorithms. In addition, the proposed algorithm always performs well in nine large-scale artificial data sets with 6,000 to 50,000 nodes and three large realistic network datasets, which verifies its computational performance and utility in community detection for large-scale networks.

## INTRODUCTION

With the rapid development of the Internet and big data technology, research on complex networks has gradually penetrated into many fields, such as information science and

Corresponding author
Jinyun Yang, yjy1129@foxmail.com

biological science, and has thus become a very challenging research topic (*Dey, Tian & Gel, 2021*). In social networks, such as scientific research cooperation and transportation networks, objects are usually represented as nodes, and relationships between objects are represented as edges (*Zhou et al., 2019*). Real-world networks have one important feature, the community structure, that is, a network is usually composed of several communities, with relatively close node connections within the community and relatively sparse node connections between the communities. The discovery of community structure is an important basis for exploring the formation principle (*Arinik, Labatut & Figueiredo, 2021*) and function of complex network structures (*Tommasel & Godoy, 2018*) and plays a vital role in many fields. For instance, in the field of biology (*Martinet et al., 2020*), community detection is of great significance for understanding the specific organizational structure, functional analysis, and behavior prediction of biological systems. In the field of e-commerce, consumers with similar purchasing habits can be mined through community detection, thus creating greater business value through the establishment of efficient recommendation systems (*Li & Zhang, 2020*). In the field of infectious diseases, community detection can be used to analyze and identify the key population of infectious diseases so as to effectively control the spread of diseases (*Chakraborty, Ghosh & Park, 2019*). Therefore, the quick and effective discovery of the community structure of networks has become the primary task and an important branch of social network research.

With the extensive research on social network analysis, many community detection algorithms have emerged, but most of them suffer from limitations such as high complexity, low accuracy of community division, and unstable results. The label propagation algorithm (LPA) has attracted attention due to its advantages of low time complexity, no prior conditions, and suitability for community detection in large-scale networks (*Li et al., 2021*). However, the traditional LPA has the following disadvantages: (1) LPA adopts a random strategy in the updating sequence of nodes, resulting in randomness in the community partition results; (2) LPA treats every node as equally important and does not distinguish the importance of each node; (3) LPA assigns a unique label to each node and fails to identify overlapping communities (*Lu et al., 2018*).

In view of the abovementioned shortcomings, numerous improved algorithms have been proposed. *Kaixuan, Hongchang & Ruiyang (2018)* proposed an improved LPA algorithm based on label propagation ability, developed a calculation method based on a k-shell decomposition algorithm for determining the importance of individual nodes (*Sun, Miao & Staab, 2021*), and formulated a label update strategy through the importance ranking of nodes and label propagation ability. *Xiaojing (2020)* proposed a community detection algorithm based on node influence and similarity (NIS-LPA), wherein the selected seed nodes are used to expand into seed regions, and then the similarity between nodes is calculated based on the network topology and real attributes of nodes, thus improving the stability and accuracy of the algorithm. *Zhenxin, Yuecheng & Yu (2021)* proposed a community detection algorithm integrating LeaderRank and label propagation (LLPA) wherein the three aspects of node label initialization, node update sequence, and label propagation selection process are improved. The LeaderRank algorithm is adopted to select key nodes, and labels are assigned to them by calculating the influence of

the nodes. Thereafter, the nodes are updated according to the influence of the nodes, and the propagation ability between nodes is considered in the process of label propagation. *Gui et al. (2018)* proposed a community detection algorithm based on boundary nodes and label propagation (LBN), which determines core nodes and boundary nodes, respectively, and then determines the community to which they belonged according to the weight of the boundary nodes, thus improving the stability of the algorithm. However, the values of $Q$ (*Yuan & Liu, 2021*) and NMI are still unsatisfactory. *Zhang et al. (2020)* proposed label importance–based label propagation algorithm (LILPA) for community detection for application in core drug detection. In LILPA, when labels are transmitted to other nodes, the label updating process based on node importance, node attractiveness, and label importance is used to improve the label instability and the accuracy and efficiency of community division. For overlapping communities, *Li et al. (2017)* proposed an efficient community detection algorithm based on label propagation with community kernel (CK-LPA), which assigns a corresponding weight to each node according to the importance of the node in the network and updates node labels according to the weight order. They also discussed the composition of weights, label updating, propagation strategies, and convergence conditions. *Jian et al. (2019)* improved the label update order and propagation threshold, and proposed an overlapping community detection algorithm by using the PageRank and node clustering coefficients algorithms (COPRAPC), wherein nodes with low influence are selected for label propagation, and the node clustering coefficient is used to control the maximum number of communities that nodes belong to *Qingshou et al. (2020)* proposed an overlapping community detection algorithm integrating label preprocessing and node influence (FLPNI), thereby greatly reducing the randomness of label propagation. *Xu, Guo & Yang (2020)* proposed an improved LPA for community detection based on two-level neighborhood similarity (TNS-LPA); defined a new two-level neighborhood similarity, which selected the initial community center by considering the minimum distance and local centrality index; and optimized the algorithm by adopting the asynchronous updating label strategy according to the importance of nodes, thereby further improving the accuracy of community division. *Li et al. (2021)* proposed an improved label propagation algorithm based on modularity and node importance (LPA-MNI) wherein the initial community is identified based on the value of modularity, and then the remaining nodes that have not been assigned to the initial community are clustered through label propagation. Node importance is used to improve the label update sequence, and the label selection mechanism is used when the majority of nodes contain multiple labels. *Kouni, Karoui & Romdhane (2020)* proposed the node importance–based label propagation algorithm (NI-LPA) to identify overlapping communities to address the problem of instability in the LPA algorithm caused by random updating. NI-LPA uses information derived from node attributes to simulate special propagation and filtering processes, and experiments revealed its high efficiency for overlapping community detection. *Wang et al. (2020)* proposed another LPA algorithm based on node importance (NI-LPA) wherein the importance of nodes is defined by combining the signal propagation of nodes, the value of K-shell nodes themselves, and the Jaccard distance between adjacent nodes, which better avoids the instability caused by

random selection of nodes in traditional LPA algorithm. *Lim, Salzman & Tsiotras (2021)* encoded both semantic and geometric information of the environment in a weighted colored graph, in which the edges were partitioned into a finite set of ordered semantic classes (*e.g.*, colors), and then incrementally searched for the shortest path among the set of paths with minimal inclusion of inferior classes (*Aghaalizadeh et al., 2021*). In the first and second iterations ($t <= 2$) of the propagation, if the number of maximum label frequencies in neighbor nodes was equal, the Adamic/Adar index was used to select the appropriate label. For the other iterations ($t > 2$), a new criterion, known as label strength, was applied to select the label with the highest strength of a node. *Zhang & Xia (2021)* proposed a new node similarity metric, and the label was updated according to the similarity between the current node and neighbor nodes.

The abovementioned algorithms focus on the calculation of the node importance and seed node selection and consider the randomness of node update order but ignore the importance of label update strategy, resulting in the unstable and less accurate community division. Therefore, this study focused on the updating strategy of nodes and labels to achieve efficient and accurate community division. The two-stage community detection algorithm based on the label propagation algorithm (*Wenping et al., 2018*) (LPA-TS) has the following problems. (1) In the first stage, the algorithm determines the node update sequence from the descending participation coefficient (PC) and then updates the node label to that with the largest similarity so as to obtain the initial partition result. However, only the number and degree of common neighbors are considered in the definition of similarity. There may be multiple nodes with maximum similarity with the same number and degree of common neighbors. If one node is randomly selected for label update, the result of community division will be unstable. (2) In the second stage, the algorithm first regards the initial community as nodes and then determines the order of community mergers from the PC. Then, the algorithm performs merging according to the conditions of a weak community and finally obtains the community structure. However, in some classical networks, the community division results are not ideal, and the modularity is low because LPA-TS has some shortcomings in the updating strategy of nodes and labels and the definition of initial community merge conditions. To solve these problems, an improved two-stage label propagation algorithm (LPA-ITSLR) was proposed in this article. The contributions and innovations of this article are as follows.

1. To solve the problem of unstable and inaccurate community division results yielded by the LPA-TS algorithm, a new similarity measurement between nodes was proposed to optimize the node label updating strategy. In the initial stage, the number and degree of common neighbors of nodes and the similarity of structural information between common neighbors are considered comprehensively. In view of the situation that multiple nodes may have the maximum similarity, the importance of nodes is sorted by calculating the LeaderRank so as to avoid the randomness of node label update order and ensure the stability of the initial community division result.

2. To address the problem of low modularity in LPA-TS, the optimal parameter value was determined by improving the definition of weak community in the original algorithm,

and the evaluation function based on complementary entropy was changed to the objective function based on modularity optimization in the community merging stage so as to further improve the quality of community division and the accuracy of the final division result.

3. Experiments were conducted on nine realistic networks and 19 artificial datasets with different scales (1,000 nodes to 50,000 nodes). The Q and NMI were used as evaluation indexes to compare the proposed algorithm with several classic algorithms. The time complexity of the algorithm was also analyzed. Experimental results showed that the improved algorithm has higher quality and stability in community division than the comparative algorithms. For large-scale data sets, the proposed algorithm can still achieve high quality of community division.

## THEORETICAL BASIS

### Community division

A complex network is generally represented by $G(V, E)$, where $V = \{v_1, v_2, \ldots, v_n\}$ is the node set, $E = \{e_1, e_2, \ldots, e_m\}$ represents the set of edges, and $\Omega = \{\Omega_1, \Omega_2, \ldots, \Omega_k\}$ is one of the division of $G$ if and only if:

1. $\forall \Omega_r \in \Omega, \Omega_r \neq \varnothing$;
2. *For all* $\Omega_r \in \Omega, \cup_{\Omega_r \in \Omega} \Omega_r = V$;
3. $\forall \Omega_p, \Omega_q \in \Omega, \Omega_p \cap \Omega_q = \varnothing$;

### Participation coefficient

The PC of node $v_i$ $PC_i$ (*Wenping et al., 2018*) is used to describe the distribution of nodes with different communities in the network edge; it is defined as Eq. (1), where $k$ is the number of communities, and $d_i$ is the degree of node $v_i$ and $d_i(\Omega_r) = \left|\{v_j | (v_i, v_j) \in E \wedge v_j \in \Omega_r\}\right|$. A high PC value indicates that the node is connected with more communities and that the node has a low degree of belonging to each community. In contrast, a low PC value indicates that the node is connected to a fewer number of communities and that the node has a high degree of belonging to each community. When community detection is performed, nodes with low PC and obvious community affiliation are selected to start traversal, which is more conducive for finding the correct community structure.

$$PC_i = 1 - \sum_{r=1}^{k} \left(\frac{d_i(\Omega_r)}{d_i}\right)^2 \tag{1}$$

### Strong and weak communities

Community structures can be strong or weak (*Wenping et al., 2018*). A strong community means that the number of links between any node in the community and the inside of the community is greater than the number of links between the node and the outside of the community. It can be defined as Eq. (2). A weak community means that the sum of the

edges of all nodes in the community and the nodes inside the community is greater than the sum of the edges of all nodes outside the community. It can be defined as Eq. (3). In general, a community should satisfy at least the character of weak community.

$$\alpha * d_i^{in}(\Omega_r) > d_i^{out}(\Omega_r), \ \forall \ i \in \Omega_r \tag{2}$$

$$\alpha * \sum_{i \in \Omega_r} d_i^{in}(\Omega_r) > \sum_{i \in \Omega_r} d_i^{out}(\Omega_r) \tag{3}$$

In Eqs. (2) and (3), $d_i^{in}(\Omega_r)$ represents the number of connected edges between the node $v_i$ in the community $\Omega_r$ and the internal nodes of $\Omega_r$, $d_i^{out}(\Omega_r)$ represents the number of connected edges between node $v_i$ in $\Omega_r$ and other nodes except $\Omega_r$. In general, $\alpha = 2$.

## LeaderRank algorithm

The LeaderRank algorithm (*Zhenxin, Yuecheng & Yu, 2021*) is used to calculate the influence of nodes in the network. A background node $v_g$ is added to the network and connected with all the nodes in the network to form a new network. The algorithm assigns 1 unit of LeaderRank (LR) value to all nodes except the background node in advance, assigns 0 unit of LR value to node $v_g$, uses Eq. (4) to calculate the LR value of node $v_i$ in each iteration, and iterates repeatedly until $v_i$ reaches a steady state and then uses Eq. (5) to divide the background nodes evenly among all the nodes.

$$LR_i(t) = \sum_{j=0}^{N} \frac{a_{ij}}{d_j} LR_j(t-1) \tag{4}$$

where $t$ is the number of iterations and $N$ is the number of nodes in the network. If there is an edge between nodes $v_i$ and $v_j$, then $a_{ij} = 1$, otherwise, $a_{ij} = 0$; $d_i$ represents the degree of node $v_i$, and $a_{ij}/d_j$ represents the probability of node $v_i$ walking to node $v_j$ randomly.

$$LR_i = LR_i(t_c) + \frac{LR_g(t_c)}{N} \tag{5}$$

where $t_c$ is the number of iterations when it reaches stability and $LR_i(t_c)$ is the LR value when node $v_i$ reaches stability. Similarly, $LR_g(t_c)$ is the $LR$ when node $v_g$ reaches stability.

## Evaluation indicators

### Modularity

Modularity ($Q$) is commonly used for measuring the strength of community structures. The closer its value is to 1, the better the quality of community division (*Yuan & Liu, 2021*). $Q$ can be calculated as follows in Eq. (6). Where $m$ is the total number of edges in the network, $a_{ij}$ is an element in the adjacency matrix $A$ of network $G$, and $d_i$ is the degree of node $v_i$. When $v_i$ and $v_j$ belong to the same community, $\delta_{i,j} = 1$; otherwise $\delta_{i,j} = 0$.

$$Q = \frac{1}{2m} \sum_{i,j} \left( a_{ij} - \frac{d_i d_j}{2m} \right) \delta_{i,j} \tag{6}$$

*Normalized mutual information*

For networks with a known community structure, NMI is generally used to evaluate the community division effect. The higher the NMI, the more similar the result is to the realistic community structure. A value of 1 indicates that the partition result is completely consistent with the actual community structure. Assuming that $A = \{A_1, \ A_2, \ \ldots, \ A_k\}$ and $B = \{B_1, \ B_2, \ \ldots, \ B_{k'}\}$ are the realistic community structure and the division result of the network by an algorithm, respectively, $k$ and $k'$ are the number of communities under the two divisions, NMI can be defined as follows in Eq. (7). Where $n$ is the total number of nodes, $T$ is the confusion matrix, $T_{ij}$ is the number of common nodes included in the realistic communities.

$A_i$ and $B_j$, and $T_i$ is the sum of the elements in the $i$-th row of the confusion matrix.

$$\text{NMI} = \frac{2\sum_{i=1}^{k}\sum_{j=1}^{k'} T_{ij}log\dfrac{nT_{ij}}{T_iT_j}}{-\sum_{i=1}^{k} T_ilog\dfrac{T_i}{n} - \sum_{j=1}^{k'} T_jlog\dfrac{T_j}{n}} \tag{7}$$

# THE PROPOSED ALGORITHM

## Question-posing

In the first stage of the LPA-TS algorithm, when there are two or more nodes with the largest similarity with the current node, the algorithm randomly selects one node for label update; this may lead to unstable partition results. LPA-TS algorithm expresses the similarity of nodes as $CN$, which can be expressed as Eq. (8), where $N_i$ represents the neighbor nodes of node $v_i$ and $d_i$ represents the degree of node $v_i$.

$$CN(v_i, \ v_j) = \left| N_i \cap N_j \right| + \frac{1}{d_id_j} \tag{8}$$

In the network diagram shown in Fig. 1, when the LPA-TS algorithm is used for the first stage of community division, two initial rough communities are obtained: $\Omega_1 = \{v_7, v_8, v_9, \ v_2\}$ and $\Omega_2 = \{v_1, v_3, \ v_4, \ v_5, \ v_6\}$. The nodes in different communities are represented by different shapes and colors in Fig. 1. At this time, node $v_0$ has not yet been merged into any community. As shown in Fig. 1, $N_0 = \{v_2, v_8, \ v_1, \ v_5\} \wedge (v_2, v_8 \in \Omega_1) \wedge (v_1, v_5 \in \Omega_2)$. According to the similarity calculation formula of the LPA-TS algorithm, the node $v_0$ has a large similarity with $v_9$ in the community $\Omega_1$, and $v_0$ has a large similarity with $v_3$ in the community $\Omega_2$. Moreover, both $v_9$ and $v_3$ have the same neighbor attributes as node $v_0$, that is, $CN(v_9, v_0) = CN(v_3, v_0)$. At this time, the LPA-TS algorithm randomly selects a community to merge $v_0$ into it and finally yields two community division results, $\Omega = \{\{v_7, v_8, \ v_9, \ v_2\}, \ \{v_0, v_1, v_3, \ v_4, \ v_5, \ v_6\}\}$ and $\Omega' = \{\{v_7, v_8, v_9, \ v_2, \ v_0\}, \ \{v_1, v_3, \ v_4, \ v_5, \ v_6\}\}$, resulting in instability.

The traditional LPA algorithm and the LPA-TS algorithm were used to conduct 100 experiments on classic karate and football networks. The corresponding module degree $Q$ (*Yuan & Liu, 2021*) of the community division results is shown in Fig. 2. Both algorithms

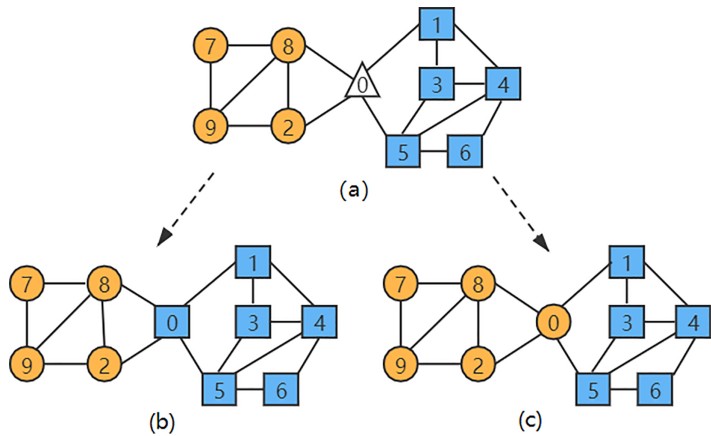

**Figure 1 (A–C) A network instance with two communities.** The nodes in different communities are represented by different shapes and colors in Fig. 1.

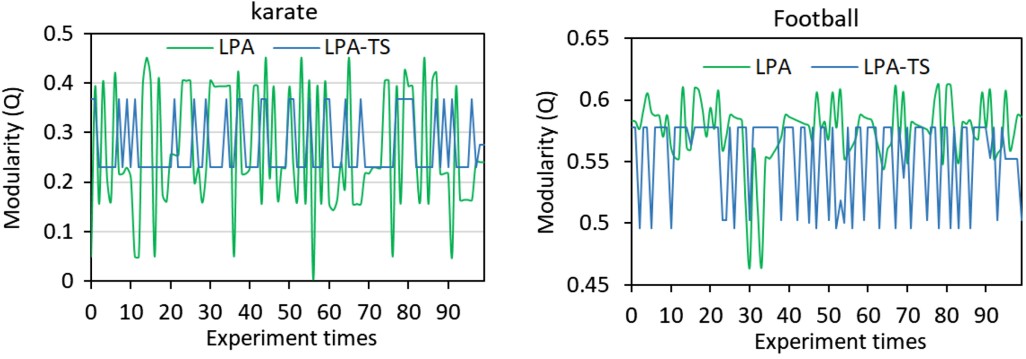

**Figure 2 Results of 100 experiments of the two algorithms on two networks.** LPA and LPA-TS were tested 100 times on two different networks.

exhibited obvious oscillations, indicating that the community division results of the algorithms are unstable.

The LPA-TS algorithm only yielded an initial community division result in the first stage, and there were still many small communities. In the Karate network shown in Fig. 3, some nodes with higher degrees have greater similarities with many nodes. For example, node $v_{34}$ can easily pass its label to neighboring nodes, while those at the edge of the network have low similarity to central nodes higher degrees. For example, nodes $v_{25}$ and $v_{26}$ can easily form small-scale communities, such as triangle nodes and diamond nodes in Fig. 3. To merge these small communities, LPA-TS uses the definition of weak communities and the evaluation function based on complementary entropy in the second stage. However, in the definition of weak communities α is set as 2, which leads to unstable division results in some networks, that is, the final community division results are not ideal, and the degree of modularity is low. Therefore, in this study, the parameters and the objective function in the second-stage community merger strategy were improved and a new community division method, LPA-ITSLR, was developed to achieve stable and more accurate community division results.

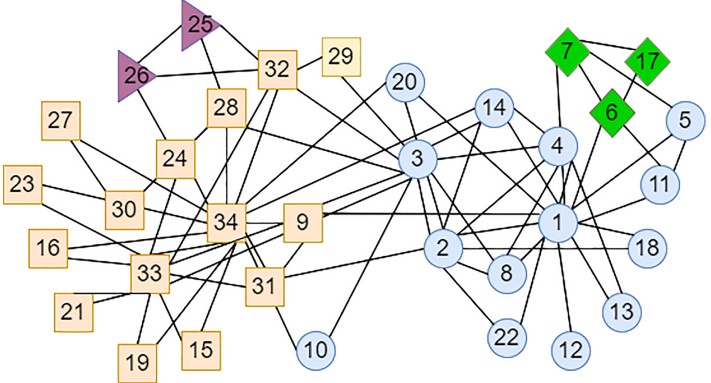

**Figure 3 LPA-TS algorithm partitioning results for Karate network in the first stage.** Different colors represent different neighborhoods.           

## Node similarity definition

To solve the abovementioned problems, a new similarity index was proposed, which considers the common neighbors, degrees of nodes, and the structural relationship between common neighbors. The more common neighbors two nodes have, the more similar they are. The higher the degree of a node, the more the number of nodes it shares its edges with, that is, the similarity of two nodes is inversely proportional to the degree of the node itself. The number of connected edges between neighboring nodes is combined to avoid multiple nodes with the same similarity as the original node. The improved similarity index can be expressed as follows:

$$S_{(i,j)}^{ICN} = \frac{|N_i \cap N_j| + 1}{|N_i \cup N_j|} + \frac{1}{d_i d_j} + \frac{C(v_i, v_j)}{|N_i \cap N_j|} \tag{9}$$

where $C(v_i, v_j)$ represents the number of edges between the common neighbor nodes of nodes $v_i$ and $v_j$. The numerator of the first term of the equation is increased by 1 so that the improved similarity index is not 0 when there is no public neighbor. According to the definition of similarity in Eq. (9), the similarity between nodes $v_0$ and $v_9$ in Fig. 1 is greater; thus, node $v_0$ is merged into the community $\Omega_1$, which solves the problem of unstable community division results that may occur in the LPA-TS algorithm.

## Algorithm description

### Stage 1: initial community detection

The first stage of the proposed LPA-ITSLR algorithm is shown in Table 1. First, each node is assigned a unique label, and the similarity between nodes is calculated according to Eq. (9). Then, the PC value of each node is calculated and sorted in ascending order. Next, the labels are updated according to the sorted nodes. In the label updating strategy, the similarity between the current node and other nodes is compared. If the node with the largest similarity is not unique, the LR is further compared; if not, one is randomly selected to obtain the rough initial community structure in the first stage.

**Table 1 First stage of the LPA-ITSLR algorithm.**

**Step 1**: Initial community discovery

**Input**: $G\ (V, E)$

**Output**: Label of each node $v_i \in V$, $L(V) = \{l_1, l_2, \ldots, l_q\}$

1: assign a unique label to each node in the $G$;

2: **for** each $v_i$, $v_j \in V$, calculate $S^{ICN}_{(i,j)}$ according to Formula (9), **end for**;

3: $U \leftarrow$ Calculate the $PC_i$ for each node according to Formula (1);

4: **Sort** ($U$) by $PC_i$ in ascending order;

5: **for** each $v_i \in U$:

6:    $S \leftarrow \arg\max S^{ICN}_{(i,j)}$,  where $v_j \in N_i$;

7:    **if** $|S| \neq 1$:

8:       $L \leftarrow \arg\max LR(v_j)$,  where $v_j \in S$

9:       **if** $|L| \neq 1$: $k = $ random $(|L|)$, $l_i \leftarrow l_k$;

10:      **else**: $l_i \leftarrow l_L$;

11:   **else**: $l_i \leftarrow l_S$;

12: **end for**

### Stage 2: community merge

The second stage of the LPA-ITSLR algorithm is shown in Table 2. In view of the problem that small communities may cause low modularity in the first stage, whether the initial community meets the weak community condition is judged first. If the condition is not met, the community with the largest number of connected edges is selected for merging; this process is repeated until the entire network meets the weak community condition. The research of LPA-TS (*Wenping et al., 2018*) shows that α is generally set as 2 in Eq. (3). In order to achieve more accurate results, α is set as 0.5, 1, 1.2, 1.5 and 2, respectively for the eight data sets used in this article. Through experiments, it can be known that when α is set as 1.5, good division results are achieved. Moreover, experiments are also carried out on 15 artificial data sets to verify the rationality of the value of α. Therefore, in our study, α is determined to be 1.5 to achieve better performance. Each community is regarded as a node, and its PC value is calculated using Eq. (1); then, the community with the most links is determined for merging. If the modularity increases after the merge, the merge will be selected; otherwise, it will not be merged, thus ensuring that the community structure after the second stage merge will have a higher modularity and be closer to the realistic community structure.

## EXPERIMENT AND ANALYSIS

In this study, numerous experiments are conducted on representative realistic networks and artificial datasets with different structural parameters. The traditional LPA algorithm, LPA-TS algorithm, and several classic community detection algorithms were compared; moreover, the effectiveness, correctness, stability, and accuracy of the proposed algorithm were verified.

**Table 2 Second stage of the LPA-ITSLR algorithm.**

**Step 2**: Community merge

**Input**: $L(V) = \{l_1, l_2, \ldots, l_q\}$

**Output**: $\Omega = \{\Omega_1, \Omega_2, \ldots, \Omega_k\}$

1:   **for** $l_i$ in $L(V)$:

2:     if $(\alpha * \sum_{j \in l_i} d_j^{in}(l_i) > \sum_{j \in l_i} d_j^{out}(l_i)) == flase$:

3:       $l_i = l_j;$  ($where$ $l_j = \max \sum_{v_i \in l_i,\ v_j \in l_j} a_{ij})$

4:   **end for**

5:   get the community set $\Omega' = \left\{\Omega_1^{(0)}, \Omega_2^{(0)}, \ldots, \Omega_k^{(0)}\right\}$, $n' = |\Omega'|$, $t = 0$;

6:   take each $\Omega_i^{(t)}$ as a node $s_i$;

7:   take the number of edges between nodes as weight;

8:   get the new network $G' = (V', E')$, where $V' = (s_1, s_2, \ldots, s_{n'})$;

9:   $t = 0$; Calculate $Q(t)$;

10: **do**:

11:    $s_m = \arg\max PC(s_i)$;

12:    $j = argmax\omega_{mj};$  $\left(where\ \omega_{mj} = \sum_{v_i \in s_m,\ v_j \in s_j} a_{ij},\ s_j \in V',\ j \neq i\right)$

13:    try merge $\Omega_m^{(t)}$ and $\Omega_j^{(t)}$ and Calculate $Q(t+1)$;

14:    if $(Q(t+1) > Q(t))$:

15:      merge $\Omega_m^{(t)}$ and $\Omega_j^{(t)}$; $t = t+1$, $n' = n' + 1$;

16: **while** $(Q(t+1) > Q(t))$

17: **return** $\Omega^{t+1}$

**Table 3 Basic structural parameters of real datasets.**

| Dataset | $|V|$ | $|E|$ | $|\Omega|$ | $max(k)$ | $<k>$ | $<d>$ | $<c>$ |
|---|---|---|---|---|---|---|---|
| Karate | 34 | 78 | 2 | 17 | 4.588 | 2.408 | 0.588 |
| Dolphin | 62 | 159 | 2 | 12 | 5.129 | — | 0.309 |
| Polbooks | 105 | 441 | 3 | 25 | 8.4 | 3.079 | 0.448 |
| Football | 115 | 613 | 12 | 12 | 10.661 | 2.508 | 0.403 |
| Les_Miserable | 77 | 254 | — | 36 | 6.597 | 2.641 | 0.736 |
| NetScience | 379 | 914 | 16 | 34 | 3.451 | 6.042 | 0.798 |

**Note:**
   Six classic real social network datasets were used in the experiment; their attribute characteristics are presented in Table 3.

## Analysis of experimental results on realistic networks

### Realistic dataset

Six classic realistic datasets were used in the experiment; their attributes are presented in Table 3. Here, $|V|$ represents the total number of nodes in the network, $|E|$ represents the total number of edges, $|\Omega|$ represents the number of communities included in the network, $max(k)$ represents the maximum node degree, $<k>$ represents the average node

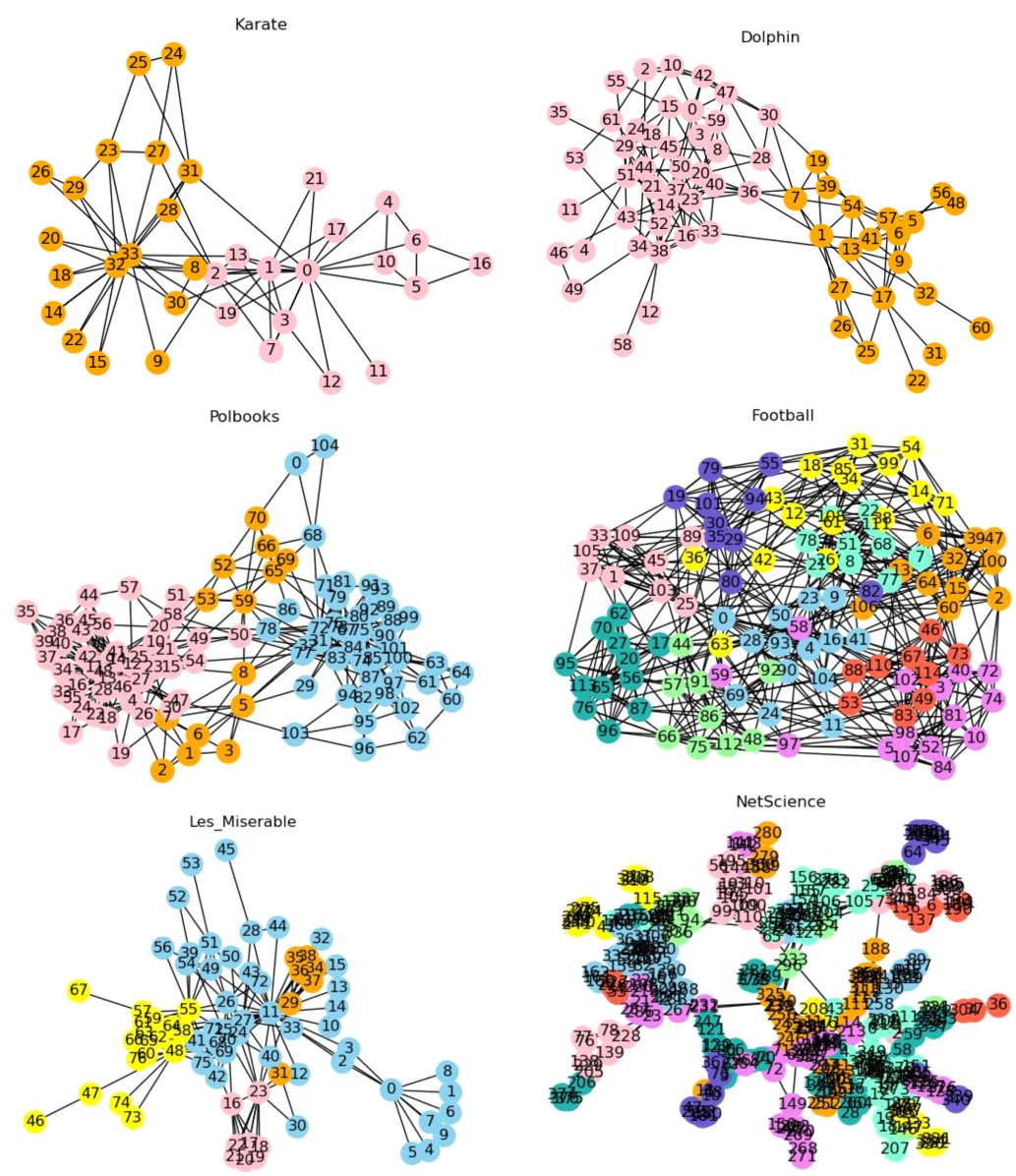

**Figure 4 Community partition results of real networks.** The partition results of this algorithm on six real data sets are shown.

degree, <*L*> represents the average path length, and <*c*> represents the clustering coefficient.

### Community division results

The proposed LPA-ITSLR algorithm was used to divide communities in the six abovementioned real datasets. The results are illustrated in Fig. 4, where nodes in different communities are represented by different color.

**Table 4  Average modularity values of 10 experiments for the three algorithms on real datasets.**

| Dataset/$\langle Q \rangle$ | LPA | LPA-TS | LPA-ITSLR |
|---|---|---|---|
| Karate | 0.3174 | 0.3716 | 0.4242 |
| Dolphin | 0.4920 | 0.3759 | 0.5418 |
| Polbooks | 0.3801 | 0.4569 | 0.5207 |
| Football | 0.5819 | 0.6010 | 0.6068 |
| Les_Miserable | 0.2719 | 0.5007 | 0.5102 |
| NetScience | 0.7769 | 0.7573 | 0.7567 |

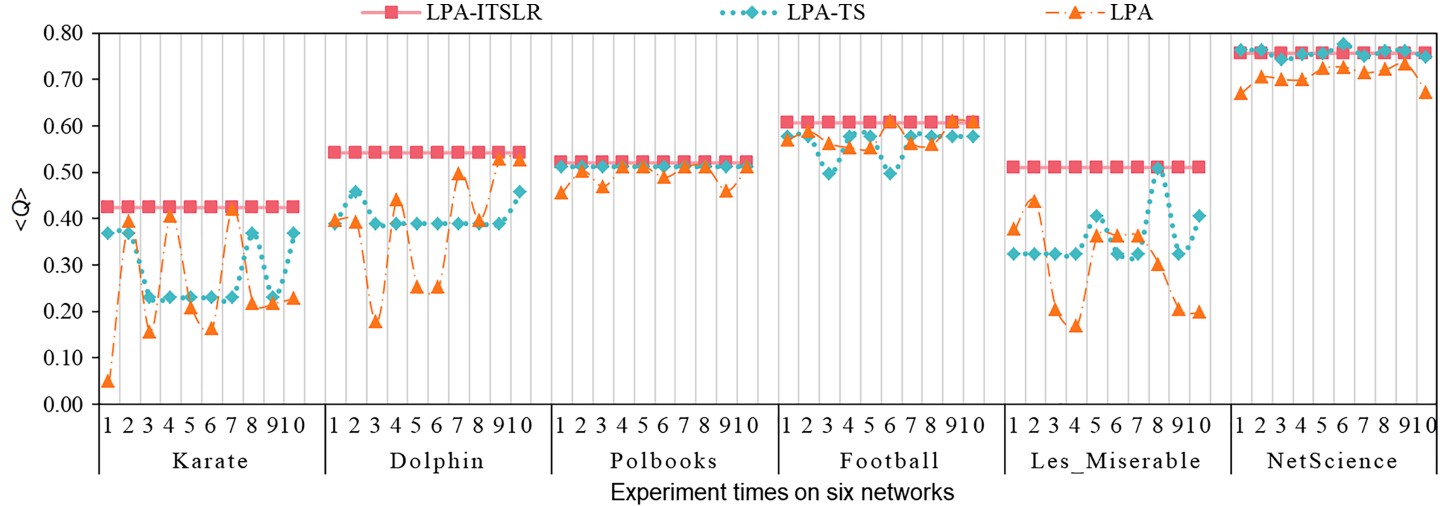

**Figure 5  Comparison of algorithm stability.** The modularity comparison of three algorithms on six networks.

### Stability analysis of LPA-ITSLR

The proposed LPA-ITSLR algorithm and the LPA and LPA-TS algorithms were compared and analyzed in the six abovementioned real datasets. Each dataset was run independently for 10 times, and the average value of the three algorithms on the six datasets was obtained (denoted as $\langle Q \rangle$), as shown in Table 4. The independent experimental results for each time are shown in Fig. 5. As can be seen from the experimental results presented in Table 4 and Fig. 5, LPA-ITSLR performed well in all datasets, with the exception that the average module degree on the NetScience was slightly lower than that obtained using the other two algorithms. Moreover, LPA-ITSLR yielded more stable community partitioning results and a higher modularity than the other two algorithms. NetScience is a weighted network; however, in the experiment, the weight was ignored, and it was transformed into a powerless network for community division. Therefore, the quality of community division on this network obtained using LPA-ITSLR was slightly lower than that obtained using the other two algorithms. However, in the 10 independent experiments, the results of the LPA and LPA-TS algorithms exhibited fluctuations, indicating that the two algorithms are unstable due to the randomness of

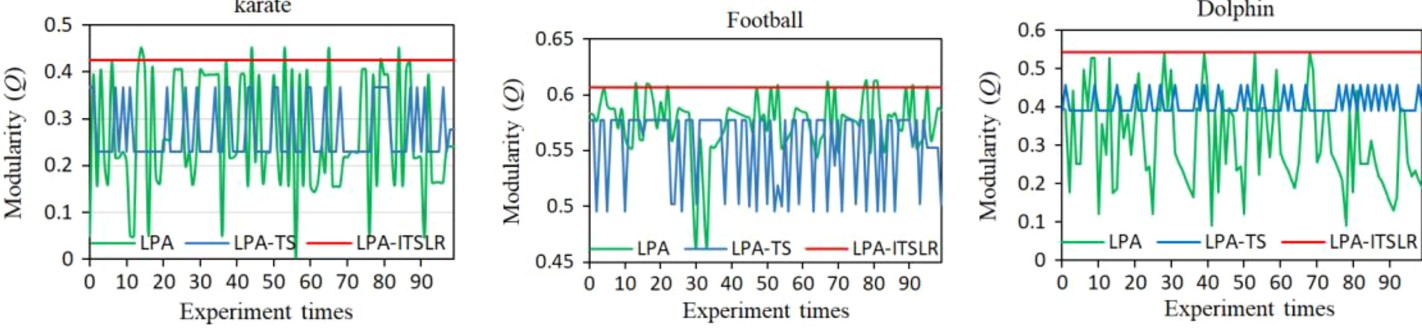

**Figure 6 Comparison of modularity of LPA, LPA-TS, and LPA-ITSLR.** The results of 100 experiments were compared on Karate, Football and Dolphin networks.

**Table 5 Modularity comparison of five algorithms.**

| Network | Karate | Dolphin | Polbooks | Football |
|---|---|---|---|---|
| COPRA | 0.2348 ± 0.10187 | 0.3741 ± 0.03946 | 0.4884 ± 0.03215 | 0.5972 ± 0.02115 |
| WLPA | 0.3682 ± 0.08176 | 0.3695 ± 0.02517 | 0.5070 ± 0.00622 | 0.5981 ± 0.01374 |
| LINSIA | 0.3989 ± 0.00004 | 0.3878 ± 0.00005 | 0.4521 ± 0.00007 | 0.5853 ± 0.00007 |
| LILPA | 0.4213 ± 0.0029 | 0.4003 ± 0.00214 | 0.4635 ± 0.00646 | 0.6061 ± 0.00151 |
| LPA-ITSLR | 0.4242 | 0.5418 | 0.5207 | 0.6068 |

node and label update. The module-degree value of the proposed LPA-ITSLR algorithm always remained stable for every network, indicating that LPA-ITSLR effectively solves the oscillation problem in the process of label propagation and has higher accuracy and stability.

To further verify the robustness of LPA-ITSLR, 100 independent experiments were conducted on the Karate, Dolphin, and Football networks; the results are presented in Fig. 6. The community division results obtained using the LPA algorithm exhibited the most serious fluctuations in the modularity value, followed by the LPA-TS algorithm. In contrast, LPA-ITSLR maintained the same community division results in 100 experiments, and the modularity was higher than that of LPA and LPA-TS.

To further evaluate the performance of the LPA-ITSLR algorithm, it was compared with four recent community detection algorithms based on label propagation. Among them, the COPRA algorithm (*Gregory, 2010*) realizes community division by assigning multiple labels with attribution coefficients to a node. The WLPA algorithm (*Tong et al., 2015*) first selects the label with a larger weight for propagation during the label propagation process. The LINSIA (*Wu et al., 2016*) algorithm is based on node importance and employs label importance to complete the community division. The LILPA (*Zhang et al., 2020*) algorithm uses a fixed label update sequence based on the ascending order of node importance. The modularity of the results obtained using the five algorithms on the four real datasets is presented in Table 5. From Table 5, it can be seen LPA-ITSLR yielded

**Table 6 Results of eight algorithms on classical networks.**

| Network | Karate | | Dolphin | | Football | |
|---|---|---|---|---|---|---|
| Criteria | $|\Omega|$ | Q | $|\Omega|$ | Q | $|\Omega|$ | Q |
| Fastgreedy | 3 | 0.38 | 4 | 0.495 | 6 | 0.549 |
| LPA | 2 | 0.292 | 3 | 0.492 | 9 | 0.576 |
| Leading Eigenvector | 4 | 0.393 | 5 | 0.491 | 8 | 0.492 |
| Walktrap | 5 | 0.353 | 4 | 0.489 | 10 | 0.602 |
| NIBLPA | 3 | 0.352 | 5 | 0.452 | 9 | 0.542 |
| EdMot | 3 | 0.412 | 4 | 0.518 | 9 | 0.604 |
| LPA-MNI | 2 | 0.372 | 4 | 0.527 | 11 | 0.582 |
| LPA-ITSLR | 2 | 0.4242 | 2 | 0.5418 | 10 | 0.6068 |

**Table 7 Description of synthetic networks.**

| Network | $|V|$ | $<k>$ | max(k) | min $|\Omega|$ | max $|\Omega|$ | $\mu$ |
|---|---|---|---|---|---|---|
| LFR-1–LFR-8 | 1,000 | 20 | 50 | 10 | 50 | 0.1–0.45 |
| LFR-9 | 5,000 | 10 | 50 | 50 | 50 | 0.1 |
| LFR-10 | 5,000 | 10 | 50 | 50 | 50 | 0.3 |

the highest modularity and most stable in community division results. Thus, instability caused by label oscillation is avoided effectively by using LPA-ITSLR.

### Performance comparison of LPA-ITSLR with other algorithms

For further analysis of the effectiveness of the proposed algorithm for community partition and correctness, three classic datasets of Karate, Dolphins, and Football were used, and the LPA-ITSLR algorithm and seven classic community detection algorithms were employed for obtaining the division results for correlation analysis in terms of the number of communities $|\Omega|$ and module Q as evaluation indicators, The results (*Newman, 2006*) are presented in Table 6 (*Li et al., 2019*; *Xing et al., 2014*).

From Table 6, it can be seen that the number of communities and modularity of the partition results of the eight algorithms on the three classical networks (*Yin et al., 2018*) were different, but LPA-ITSLR exhibited good performance on these datasets; moreover, the partition results and the number of communities were consistent with the realistic network structure, and the modularity was higher than that obtained using other algorithms.

### Analysis of experimental results of artificial datasets

#### Artificial datasets

Ten artificial networks were generated using the LFR benchmark (*Yang, Perotti & Tessone, 2017*); the basic information is presented in Table 7. The number of nodes $|V|$ in the top eight artificial networks is 1,000, and the community size $|\Omega|$ is 10–50, that is, $\min|\Omega| = 10$, $\max|\Omega| = 50$. The average degree of nodes $<k>$ is 20, and the maximum degree max(k)

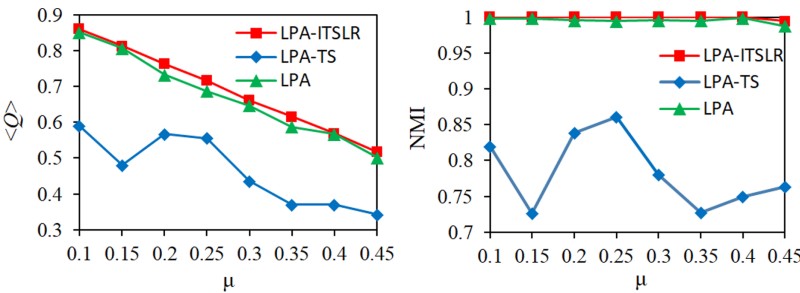

**Figure 7 Comparison of modularity and NMI on eight synthetic datasets.** The modularity and NMI results of the three algorithms under different μ values were compared.

is 50. The values of 0.1, 0.15, 0.2, 0.25, 0.3, 0.35, 0.4, and 0.45 were employed as the mixing parameter μ, and the eight networks were denoted as LFR-1–LFR-8. The latter two artificial networks are more complicated. The number of nodes is 5,000, the community size is 50, the average degree of nodes is 10, and the maximum degree is 50. The mixing parameter μ was 0.1 and 0.3, respectively, and these two networks were denoted as LFR-9 and LFR-10.

### Comparative analysis of algorithm performance

For the first eight artificial datasets, the proposed LPA-ITSLR algorithm was compared with the LPA and LPA-TS algorithms in terms of the community division results. The average modularity $<Q>$ and NMI were used as evaluation indicators. The experimental results are shown in Fig. 7. As the value of μ increased, the network became more complex. The modularity of the community division results of the three algorithms on the corresponding network decreased by varying degrees, but LPA-ITSLR yielded higher modularity than the other algorithms. Moreover, the NMI value of LPA-ITSLR on the first seven networks was 1, and the NMI value of the network with a μ value of 0.45 was 0.9943, showing extremely strong stability and higher quality of community division.

For large-scale artificial networks LFR-9 and LFR-10 with high complexity, the proposed algorithm was compared with seven recent LPA algorithms. *Q* and *NMI* were considered as evaluation parameters. The results are presented in Table 8. It can be seen that the community division results (*Liu et al., 2015*) obtained using the seven algorithms (*Xie, Szymanski & Liu, 2011*) were not stable, and the algorithm proposed in this article maintains stable community division results on the two complex artificial data sets (*Zhang et al., 2017*). Although the NMI value was slightly lower than that for other algorithms, the modularity was far higher. In the community merge phase optimization strategy based on modularity, LPA-ITSLR is superior as it can yield stable and high-quality community division results.

For the above 10 artificial networks, the experimental results show that the proposed algorithm is superior to other algorithms in both *Q* and NMI. In order to further verify the superiority of the proposed algorithm, we compare the number of communities detected by LPA-ITSLR algorithm with the actual number of communities of the 10 networks, and the results are shown in Table 9. It can be seen from Table 9 that the number

**Table 8 Results for LFR9 and LFR10.**

| Criteria | NMI | | Q | |
|---|---|---|---|---|
| Network | LFR-9 | LFR-10 | LFR-9 | LFR-10 |
| COPRA | 0.9853 ± 0.00466 | 0.9859 ± 0.00413 | 0.4259 ± 0.00786 | 0.3353 ± 0.00286 |
| SLPA | 0.9994 ± 0.00081 | 0.9931 ± 0.00352 | 0.4467 ± 0.00122 | 0.3437 ± 0.00193 |
| LINSIA | 0.8813 ± 0.00000 | 0.8267 ± 0.00007 | 0.3221 ± 0.00007 | 0.3107 ± 0.00007 |
| DLPA+ | 0.9887 ± 0.00135 | 0.9414 ± 0.00156 | 0.4423 ± 0.00164 | 0.3381 ± 0.00074 |
| WLPA | 0.9980 ± 0.00113 | 0.9979 ± 0.00111 | 0.4443 ± 0.00174 | 0.3366 ± 0.00145 |
| LPA_NI | 0.9987 ± 0.00082 | 0.9847 ± 0.00124 | 0.4467 ± 0.00024 | 0.3437 ± 0.00112 |
| LILPA | 0.9955 ± 0.00084 | 0.9692 ± 0.00115 | 0.4472 ± 0.00011 | 0.3453 ± 0.00041 |
| LPA-ITSLR | 0.9862 | 0.9531 | 0.8782 | 0.8091 |

Note:
For large-scale artificial networks LFR-9 and LFR-10 with high complexity, the proposed LPA-ITSLR algorithm was compared with seven recent label propagation algorithms for community division. $Q$ and *NMI* were considered as evaluation parameters.

**Table 9 Actual number of communities and the number of communities detected by LPA-ITSLR.**

| LFR networks | Actual number of communities | Number of communities divided by LPA-ITSLR |
|---|---|---|
| LFR-1 | 35 | 40 |
| LFR-2 | 35 | 35 |
| LFR-3 | 38 | 38 |
| LFR-4 | 45 | 45 |
| LFR-5 | 39 | 39 |
| LFR-6 | 42 | 42 |
| LFR-7 | 42 | 42 |
| LFR-8 | 42 | 40 |
| LFR-9 | 85 | 81 |
| LFR-10 | 98 | 69 |

Note:
The number of real communities and algorithm division.

of communities detected by the algorithm proposed in this article is basically consistent with the actual number of communities. In general, good results are obtained except for small deviations in some networks.

## Performance analysis of the algorithm for large data sets
### *Experiments on large scale artificial datasets*

In order to further verify the effectiveness, the computational performance and utility of the proposed algorithm for large-scale networks, nine artificial data sets were used and the number of nodes of these networks was from 6,000 to 50,000, and these networks were denoted as LFR-11 to LFR-19, respectively. Table 10 shows the experimental results on these nine large-scale networks, including the actual number of communities, the number of communities detected by the LPA-ITSLR algorithm, $Q$ and NMI.

It can be seen from Table 10 that, the algorithm performs well on these large data sets. With the increase of the number of nodes, the network scale and complexity continues to

**Table 10 Community detection results of nine large-scale artificial networks.**

| Dataset | $\|V\|$ | $<k>$ | $max(k)$ | $min\,\|\Omega\|$ | $max\,\|\Omega\|$ | $\mu$ | Actual number of communities | Number of communities found | $<Q>$ | NMI |
|---------|---------|-------|----------|------------|------------|-------|------------------------------|------------------------------|-------|-----|
| LFR-11 | 6,000 | 10 | 50 | 30 | 60 | 0.1 | 125 | 128 | 0.8730 | 0.9762 |
| LFR-12 | 7,000 | 10 | 50 | 30 | 60 | 0.1 | 130 | 133 | 0.8686 | 0.9510 |
| LFR-13 | 8,000 | 10 | 50 | 30 | 60 | 0.1 | 176 | 176 | 0.8828 | 0.9805 |
| LFR-14 | 9,000 | 10 | 50 | 30 | 60 | 0.1 | 175 | 178 | 0.8722 | 0.9629 |
| LFR-15 | 10,000 | 10 | 50 | 30 | 60 | 0.1 | 175 | 180 | 0.8775 | 0.9678 |
| LFR-16 | 20,000 | 10 | 50 | 30 | 60 | 0.1 | 436 | 464 | 0.8842 | 0.9844 |
| LFR-17 | 30,000 | 10 | 50 | 30 | 60 | 0.1 | 668 | 683 | 0.8846 | 0.9851 |
| LFR-18 | 40,000 | 10 | 50 | 30 | 60 | 0.1 | 1,058 | 1,049 | 0.8837 | 0.9793 |
| LFR-19 | 50,000 | 10 | 50 | 30 | 60 | 0.1 | 1,382 | 1,341 | 0.8643 | 0.9637 |

**Note:**
Large scale data results presentation.

expand, and there is a discrepancy between the actual number of communities and the number of communities obtained by the algorithm. But the Q is always above 0.86, and the NMI is more than 0.96 by and large. Specially, on the dataset containing 20,000 to 50,000 nodes, the NMI basically reaches more than 0.98, which shows the utility of the proposed algorithm in community division for large-scale networks. In addition, the number of communities detected by the algorithm proposed is basically consistent with the actual number of communities, which further verifies the effectiveness and superiority of the LPA-ITSLR algorithm.

### Experiments on large scale realistic datasets

Similarly, experiments were also carried out on large-scale realistic data sets to further verify the performance of the proposed algorithm. There are a variety of community discovery algorithms, mainly divided into split-based methods, such as GN (Girvan-Newman) algorithm (*Poggiolini, 2012*); methods based on modularity, such as the CNM (Clauset-Newman Modularity) algorithm (*Clauset, Newman & Moore, 2004*); methods based on spectral analysis, such as SC (Spectral Clustering) (*Kumar et al., 2015*), and methods based on label propagation, such as LPA. In addition to the six classic small realistic data sets mentioned above, three representative large-scale realistic networks were obtained, and comparative experiments were conducted with the four classic community discovery algorithms mentioned above. The three data sets are Email, Political Blogs (PB) and Power Grid (PG), and the topological properties are shown in Table 11.

The community discovery in realistic networks is more challenging than that in the simulation network, and the community structure cannot be predicted in advance, so the modularity can only be used for comparison. Table 12 shows the community division results of the proposed algorithm and the four classic algorithms on the above three large-scale realistic networks. In Table 12, the first column is a list of realistic networks, and the second to sixth columns are five classic community discovery algorithms. For each algorithm, the maximum modularity and the number of communities are computed. For example, in the Email network, the maximum modularity obtained by GN algorithm is 0.532, and the number of communities founded is 61, which is recorded as 0.532/61.

**Table 11 Properties of large-scale social network topology.**

| Network | $|V|$ | $|E|$ | $max(k)$ | $<k>$ | $<d>$ | $<c>$ |
|---|---|---|---|---|---|---|
| Email | 1,133 | 5,451 | 71 | 9.6220 | 3.6060 | 0.2540 |
| PB | 1,224 | 33,430 | 702 | 54.6242 | 3 | 0.2259 |
| PG | 4,941 | 6,594 | 19 | 2.6691 | 20.0941 | 0.1031 |

Note:
Large scale realistic data set parameter display.

**Table 12 Comparison of community division results of five classic algorithm.**

| Network | GN | CNM | SC | LPA | LPA-ITSLR |
|---|---|---|---|---|---|
| Email | **0.532**/61 | 0.446/10 | 0.412/45 | 0.014/4 | 0.504/7 |
| PB | 0.418/205 | 0.426/77 | 0.328/62 | 0.410/3 | **0.428**/2 |
| PG | 0.857/39 | **0.934**/42 | 0.830/42 | 0.871/38 | 0.931/42 |

Note:
The data with the largest modularity value in the table is displayed in bold font, and the data with the second largest value is underlined.

It can be seen from Table 12 that the algorithm proposed in this article has achieved better community division results on the three data sets in terms of modularity. On the Email network, the modularity is slightly lower than GN algorithm. On the PB network, the performance is better than the other four algorithms. On the PG network, the modularity of the proposed algorithm is only 0.003 lower than that of CNM algorithm. In general, the GN algorithm based on global, the CNM algorithm considering modularity increment and the LPA-ITSLR algorithm proposed in this article are not very different from each other in community division results for the three datasets. However, GN and CNM algorithms have higher computational complexity than other algorithms, while SC algorithm and LPA algorithm perform relatively poor. From the perspective of the number of communities, the GN algorithm tends to get more communities. For example, the GN algorithm divides the PB network into 205 communities, which is significantly higher than other algorithms. The LPA algorithm divides the PB network into three communities, which is closest to the realistic number of communities. For PB network, the algorithm proposed in this article achieves a result that is completely consistent with the number of realistic communities, while the number of communities given by other methods is more than 10. Obviously, the corresponding methods tend to over fit. Combining the two indicators of modularity and the number of communities, experimental results on six small-scale and three large-scale realistic data sets show that the LPA-ITSLR algorithm proposed can effectively realize good community division with a higher modularity, and the number of communities discovered is basic consistent with the realistic community structure.

## Comparison of time complexity of algorithms

LPA is a fast and nearly linear time-complexity algorithm for community discovery. However, the traditional LPA algorithm has poor stability due to the randomness of node selection and label update. Therefore, this study improved the LPA algorithm and proposed the LPA-ITSLR algorithm. For the algorithm proposed in this article, in the first

**Table 13 Comparative analysis of time complexity of algorithms.**

| Algorithm | Time complexity |
|---|---|
| GN | $O(nm^2)$ |
| Newman Fastgreedy | $O(n(m+n))$ |
| Edge-Betweenness | $O(m^2n)$ |
| CNM | $O(n(\log n)^2)$ |
| SC | $O(mKt+nK^2t+K^3t+n^3)$ |
| Walktrap | $O(n^2m)$ |
| LPA | $O(m)$ |
| NIBLPA | $O(m)$ |
| LPA-MNI | $O(m+n\log n)$ |
| LPA-TS | $O(n^2+c^2+t(n+c))$ |
| LPA-ITSLR | $O(n^2+n\log n)$ |

stage, the similarity between nodes is firstly calculated, and the time complexity is $O(nk)$, where $n$ is the number of nodes in the network and $k$ is the average degree of nodes. The time complexity corresponding to the computing of PC values is $O(n^2)$. Quicksort is used to determine the node update sequence, and the corresponding time complexity is $O(n\log n)$. The time complexity of the label propagation is $O(n)$. In the second stage, the algorithm first judges whether the communities generated in the first stage meet the conditions of the weak community, which takes $O(c)$ time complexity, where $c$ is the number of communities formed in the first stage, so $c$ is far less than $n$. The time complexity of the later stage of community merging is $O(c^2)$. So, the computational complexity of the proposed algorithm in this article is $O(nk)+O(n^2)+O(n\log n)+O(n)+O(c)+O(c^2)$, which is approximately equal to $O(n^2+n\log n)$. Table 13 lists the time complexity analysis results of several classic community discovery algorithms.

In the Table 13, $n$ represents the number of nodes in the network, $m$ represents the number of edges, $K$ is the number of eigenvectors, $t$ represents the number of iterations of the algorithm, and $d$ represents the depth of the tree. The first seven algorithms in Table 13 are classic community discovery algorithms, while the last four are community discovery methods based on label propagation proposed in recent years. Among them, the GN algorithm and Edge-betweenness algorithm are community discovery algorithms based on hierarchical clustering and splitting, respectively. Their ideas are very intuitive and the effect is good. However, the Edge-betweenness algorithm needs to repeatedly compute the shortest path, so the time complexity is high. The Fastgreedy algorithm is a community discovery algorithm based on the idea of modularity. The CNM algorithm is a new greedy algorithm based on Newman FastGN algorithm, using the data structure of the heap to calculate and update the network modularity, which has improved the time complexity. The clustering effect of the SC algorithm depends on the similarity matrix, and the final clustering effect obtained by different similarity matrix may be very different, and the calculation complexity of the algorithm is high. The Walktrap is a community discovery method based on random walk. Due to the complexity of the loss function, the

time complexity of the Walktrap algorithm is also high (*Luo & Wu, 2021*). By comparison, the algorithm proposed in this article also carries out community merging based on modularity in the last stage, and its computational complexity is almost the same as that of the Fastgreedy algorithm. By using sparse adjacency matrix, large networks containing millions of nodes and edges can be analyzed, and better community division results can be obtained. Meanwhile, compared with LPA, the LPA-ITSLR algorithm has significantly improved the stability of community discovery results. In addition, although the time complexity of the NIBLPA algorithm is linear, and the time complexity of the LPA-ITSLR algorithm is slightly higher than that of the LPA-NMI algorithm, the proposed algorithm solves the problems of the traditional LPA algorithm and gets higher quality community division results and has better stability. On the whole, though the performance of the proposed algorithm on a certain data set is slightly inferior to other algorithms, the experimental results on nine realistic data sets and 19 simulation data sets show that the modularity and NMI of the proposed algorithm are higher than other comparison algorithms, which shows good quality and stability of community division.

## CONCLUSION

To solve the problem of unstable results and low modularity of the LPA-TS algorithm in community detection, an improved LeaderRank-based two-stage label propagation algorithm named LPA-ITSLR was proposed in this study. In the first stage, the order of node updating is determined by descending order of the PC values. In the label propagation strategy, the improved similarity index is used, and then the influence of the nodes is compared so as to obtain the initial community division. In the second stage, the community is regarded as a node, and the PC is calculated again and sorted in ascending order. For determining the optimal parameter value in the weak community condition, the community is merged. Finally, the community structure is further improved based on the modularity optimization, and the final community division result is obtained. The proposed LPA-ITSLR algorithm solves the problem that the randomness of LPA-TS algorithm may yield unstable community partition results. Moreover, LPA-ITSLR yielded higher modularity than other algorithms on nine realistic networks and 19 artificial datasets and achieved a more stable community division. However, it has a higher time complexity in the case of certain large-scale networks with special structures such as when the network community structure is complex, when there are many small communities and less contact between communities, and for nonequilibrium size distribution networks. So a community detection method based on label propagation integrated deep learning and optimization could be employed to determine the node similarities and label influence. In the future, community detection in large-scale networks will be further studied to reduce the time complexity of the algorithm, and to achieve more accurate and efficient community detection results.

### Funding

This work was supported by the National Natural Science Foundation of China (42002138, 62172352), the Natural Science Foundation of Heilongjiang Province (LH2019F042), the postdoctoral scientific research development fund of Heilongjiang Province (No. LBH-Q20073) and the Excellent Young and Middle-aged Innovative Team Cultivation Foundation of Northeast Petroleum University (KYCXTDQ202101). The funders had no role in study design, data collection and analysis, decision to publish, or preparation of the manuscript.

### Grant Disclosures

The following grant information was disclosed by the authors:
National Natural Science Foundation of China: 42002138, 62172352.
Natural Science Foundation of Heilongjiang Province: LH2019F042.
Postdoctoral Scientific Research Development Fund of Heilongjiang Province: LBH-Q20073.
Excellent Young and Middle-aged Innovative Team Cultivation Foundation of Northeast Petroleum University: KYCXTDQ202101.

### Competing Interests

The authors declare that they have no competing interests.

### Author Contributions

- Miaomiao Liu conceived and designed the experiments, analyzed the data, authored or reviewed drafts of the paper, and approved the final draft.
- Jinyun Yang conceived and designed the experiments, performed the experiments, analyzed the data, performed the computation work, prepared figures and/or tables, and approved the final draft.
- Jingfeng Guo conceived and designed the experiments, authored or reviewed drafts of the paper, and approved the final draft.
- Jing Chen conceived and designed the experiments, prepared figures and/or tables, and approved the final draft.
- Yongsheng Zhang conceived and designed the experiments, prepared figures and/or tables, and approved the final draft.

### Data Availability

The data is available at Stanford Large Network Dataset Collection: https://snap.stanford.edu/data/.

## Supplemental Information

Supplemental information for this article can be found online at http://dx.doi.org/10.7717/peerj-cs.981#supplemental-information.

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
