# Peer review of "An improved two-stage label propagation algorithm based on LeaderRank"

_PeerJ Computer Science, doi:10.7717/peerj-cs.981_

## Round 0.1 · original submission · Major Revisions

The paper is potentially publishable. I encourage the authors to undertake a revision. Please also provide a detailed response letter. Thanks.

·

Basic reporting

Line 87: The "Q" should be italicized.
Line 147: "definition similarity": Did you mean "definition of similarity"?
Line 216: Suggest to remove the word "real".

Experimental design

Line 297: Is the alpha optimized for the networks in this research work, or it is an optimized value that can be used in any other network? Kindly clarify.

Validity of the findings

Table 5 and 6: Is there a reason why the LPA-MNI is included in Table 6 but not in Table 5? As it is also one of the latest LPA variants.
Line 380: In most cases, LPA and its variants will start to deteriorate past a certain threshold of mixing parameter. I wonder how well the proposed method handles higher values of mixing parameter. For example, mu = 0.6 and 0.7.
The proposed method produces high-quality community division in terms of NMI and Q for the LFR networks. It outperformed the other methods in terms of Q (more than double in some cases). Thus, it will be interesting to see what is the number of communities that are detected by the proposed method as compared to the actual number of communities in the LFR networks.

Reviewer 2 ·

Basic reporting

This paper proposes a new community detection method based on LeaderRank and LPA, the problem has significant importance, and the proposed method seems somewhat interesting. The related work section should include more related and recent studies to state the new contribution of the proposed method against ones. I could find a few very recent studies on the use of LPA in community detection which have not been indicated; moreover, the new contributions and shortcomings of existing methods may also be included somewhere in the manuscript? A working example to illustrate the functioning of the proposed method could be of immense help to the readers.

Experimental design

The authors may consider incorporating the convergence, sensitivity to the starting conditions, and other relevant characteristics of the proposed algorithm (including time complexity analysis). The authors may compare their approach with the current benchmarks on community detection using relatively real-world data sets of significant size based on the actual CPU time (current datasets considered are too small).

Validity of the findings

The proposed method should be competitive in computational performance for large-scale network data sets (>30000 nodes, for example); otherwise, the utility of the proposed approach to deal with realistically large networks shall remain unknown.

·

Basic reporting

no comment

Experimental design

no comment

Validity of the findings

no comment

Additional comments

In this manuscript entitled “an improved two stage label propagation algorithm based on LeaderRank”, authors propose an improved two-stage label propagation algorithm to solve the problems of poor stability and low modularity.

In the introduction, authors sum up their contributions and experimental findings. They focus their study on label propagation algorithm for community detection, they point out its advantages as low time complexity and suitability for large scale-networks as well as its limitations as randomness and instability. Authors give then a short overview of improved algorithms that take into consideration the importance of each node in order to seed node selection. They detailed the LPA-TS algorithm and underlined the inaccurate proposed similarity measurement between nodes that aims to rank nodes for labeling.
They propose a new version of LPA-TS named LPA-ITSLR where an improvement of the similarity measurement is proposed as well as an optimised strategy for node label updating based on LeaderRank value, this will avoid the randomness of ordered labels update.
The paper is well organised in two main other parts beside the introduction and the conclusion.
In the second part, theoretical basis with the necessary notions and indicators are exposed and defined. Likewise, the proposed algorithm is detailed.

The third part detailed experiment and analysis, experiments are carried out on six classical social network data sets where users show that LPA_ITSLR performed well for all these data sets and yielded more stable community partitioning in comparison with LPA and LPA-TS. Performance comparison of LPA_ITSLR with four other recent community detection algorithms are also carried out on the same classical data sets. Two of these algorithms are based on the importance of nodes. Experimental results show that LPA_ITSLR yielded the highest modularity and the most stable results. Other comparison results with seven other classical community detection algorithms show also good performance especially in term of modularity on the three following datasets: Karate, Dolphin and football.
Other experiment studies were carried out on artificial sets and showed the same improvement.

Here are some suggestion and remarks:
The complexity of the algorithm LPA-ITSLR is not studied neither experimented on large-scale graph. In fact, considering the new similarity measurement as well as the new proposed strategy for updating labels leads certainly to improve the randomness and the instability of the algorithm, but may, on the other hand, slow down the processing in comparison to the classical LPA. Could you please comment this or mention it is as a perspective of your study?

On the other hand, here are some remarks related to terminology, authors use the terms “community discovery” instead of the most common used one which is “community detection”. Is there any reason to this?
In table 3, it seems to me that d is the diameter and not the path average length. Likewise, c is called “clustering coefficient” and not the aggregating one.
At line 49 the place of the citation number 1 is inappropriate, could you please put it at the end of the sentence? At line 241, please check the sentence: “the node .. and the community .. has a large similarity”

---

## Round 0.2 · Major Revisions

A further revision is needed. Please address the comments and provide a detailed response letter. Thanks.

·

Basic reporting

No comment.

Experimental design

No comment.

Validity of the findings

No comment.

Additional comments

The paper is well written and I am looking forward to any related future work on this method.

Reviewer 2 ·

Basic reporting

Supplementary material contains .class files (unreadable) of most of the code/ material, which is not following the Peerj CS Policy where code should be made available ( Though it may be with restricted access).

Experimental design

The current utility of the proposed method to deal with realistically large networks remains unknown.

Validity of the findings

I appreciate the efforts made by the authors to address my comments in the previous round (although not all of them have been answered). Different parts of the paper have been improved. However, I still have a major concern that results from previous remarks that have not been addressed.
Specifically, my comment on the "Validity of Findings" has not been satisfactorily answered. I reiterate my comment below.
"...Validity of the findings
The proposed method should be competitive in computational performance for large-scale network data sets (>30000 nodes, for example); otherwise, the utility of the proposed approach to deal with realistically large networks shall remain unknown....".

Based on the availability of large-scale network data set, one would expect a new community detection algorithm to be competitive in terms of computational performance. The authors should explain how the proposed method is competitive for realistically large networks?

---

## Round 0.3 · Major Revisions

A major revision is needed. Please address the concerns raised by the reviewers and provide a detailed response letter with tracked changes. Thanks.

Reviewer 2 ·

Basic reporting

Different parts of the paper have been significantly improved in the previous rounds of review.

Experimental design

I appreciate the author's effort to address my comments in the previous round, although not all of them have been answered. Additional experiments may be required to address my concerns which have not been addressed adequately.

Validity of the findings

I still have a major concern that has not been addressed. It is important to show the computational competitiveness of the proposed method for realistically large networks and not for LFRs (which are used by the authors to address my comments). Can the proposed method outperform any one of the benchmarks like the Fast greedy algorithm, Walktrap algorithm, or Edge Betweenness algorithm on realistically large networks? It is not easy to understand the novelty of publication without having a clear and convincing contribution, as suggested above.

·

Basic reporting

no comment

Experimental design

no comment

Validity of the findings

no comment

Additional comments

In this manuscript entitled “an improved two stage label propagation algorithm based on LeaderRank”, authors propose an improved two-stage label propagation algorithm to solve the problems of poor stability and low modularity.

According to first round review suggestions, authors improved the algorithm and carry out experiments on large-scale networks. The largest data set in the original paper contains 10000 nodes. they generated simulation data sets containing 20000, 30000, 40000 and 50000 nodes respectively. From the experimental results, they draw the conclusion that, as the size of the data set increases, the time complexity of this algorithm increases, but it can still achieve a high modularity and NMI in terms of large-scale data sets. It would be preferable to give some measurement allowing to observe how the time complexity is increasing.

---

## Round 0.4 · accepted · Accept

The paper can be accepted. Congratulations.

Reviewer 2 ·

Basic reporting

Seems ok

Experimental design

Seems ok

Validity of the findings

The authors have made efforts to address some of the major concerns raised in the previous rounds of review; I recommend this revised manuscript for further processing.

Additional comments

Authors should check leftover grammatical/typographical issues (like in Table 12) at their end.